# AG-SLAM: ACTIVE GAUSSIAN SPLATTING SLAM

## ABSTRACT

We present AG-SLAM, the first active SLAM system utilizing 3D Gaussian Splatting (3DGS) for online scene reconstruction. In recent years, radiance field scene representations, including 3DGS have been widely used in SLAM and exploration, but actively planning trajectories for robotic exploration is still unvisited. In particular, many exploration methods assume precise localization and thus do not mitigate the significant risk of constructing a trajectory, which is difficult for a SLAM system to operate on. This can cause camera tracking failure and lead to failures in real-world robotic applications. Our method leverages Fisher Information to balance the dual objectives of maximizing the information gain for the environment while minimizing the cost of localization errors. Experiments conducted on the Gibson and Habitat-Matterport 3D datasets demonstrate state-of-the-art results of the proposed method.

## 1 INTRODUCTION

Being able to autonomously explore and map an environment while localizing within that map is a core skill for a mobile robot. This problem is known as active Simultaneous Localization and Mapping (active SLAM). Active SLAM lies at the intersection of exploration and SLAM, although it introduces challenges that are not present in either – that is, a trade-off between exploration and reducing the uncertainty of the estimated state which includes the agent's pose and the map of the environment. Classical active SLAM methods often define objectives to reduce the uncertainty of the state estimate based on its covariance [10; 40; 3; 55; 5]. The covariance is readily available for classical SLAM systems which use filters to update the state estimate. However, many recent visual SLAM systems instead use a non-linear rendering loss to update the state estimate, which makes the covariance difficult to obtain. In particular, systems of this type using 3D Gaussian Splatting (3DGS) [21] for the scene representation have been developed [30; 59; 20; 16] which allow for high-fidelity rendering of novel views of the scene. In addition to cases where the reconstruction of a scene is desired for its own sake, a 3DGS scene representation extended to support open-vocabulary semantic segmentation [67; 46; 38; 63] and can be used as a basis for language-specified robotics tasks, for example 3DGS have already been used for mobile manipulation [14]. Many existing methods for such tasks currently rely on pre-scanning the scene [42; 26; 15], so the ability to efficiently and autonomously create a 3DGS representation of the scene can support work using 3DGS for these tasks. While an existing active SLAM algorithm could be used to construct a 3DGS representation using the actions and estimated poses as inputs this is less efficient and, we argue, less effective than an active SLAM system which is specifically designed for a 3DGS scene representation. We thus present the first 3DGS-based active SLAM system, allowing us to autonomously create a scene representation of a novel environment from which we can render high-fidelity color and depth images.

Classical approaches such as frontier-based exploration and A* algorithms are still used in active SLAM systems for their efficiency and simplicity [47; 2; 53; 49; 22]. However, algorithms that use only simple heuristics cannot determine the information gain, limiting their objective to simply improving coverage or having minimal travel distances. We propose to formulate this problem as an active learning problem, use heuristic approaches to efficiently generate a large number of feasible paths as candidates, and employ an uncertainty-aware algorithm to determine the best path for both localization and mapping. This allows us to balance the dual objectives of exploration and localization uncertainty reduction by using frontiers and expected information gain to drive exploration and our novel path selection algorithm to minimize the uncertainty of the state estimate. There have been previous methods for quantifying the uncertainty of radiance fields for reconstructing scenes from

given data [44; 45; 50; 12], active view selection [34; 19; 50; 25] and active reconstruction [60] or mapping [61] of scenes with given localization, and for active SLAM of small scenes with an inward-facing camera [64]. However, all the prior methods only model the uncertainty of the scene representation, whereas we also model the localization uncertainty. In addition, we consider uncertainty over paths not only single views.

To validate our approach, we evaluate our method on scenes from the Gibson [57] and Habitat-Matterport 3D [39] dataset quantitatively and qualitatively. We show superior reconstruction quality in various metrics compared to several baselines and recent state-of-the-art methods. In particular, we compare to Active Neural SLAM [4], ExplORB [36], UPEN [11], active-INR [61] and frontier-based exploration [58] using the ratio of the frontier area to the distance to it as the selection criteria. In all cases, to make a fair comparison of the rendering quality, we only use the method to select actions and keep the SLAM backend, which is used for the final rendering evaluation, the same. Our contributions can be summarized as follows:

- We propose AG-SLAM, an active SLAM system that uses a 3D Gaussian representation. To the best of our knowledge, we are the first to study active SLAM problems with a 3D Gaussian representation.
- We derive an objective function for paths in our 3D Gaussian representation that effectively balances the information gain for exploration and the cost of possible localization errors.

## 2 RELATED WORK

**Visual SLAM**    Simultaneous Localization and Mapping (SLAM) is the problem of constructing a map of an unknown environment while localizing within that map. In visual SLAM, this is done only based on visual observations. Visual SLAM systems can be categorized as either dense or sparse. Sparse methods [32; 8; 31; 1] use and reconstruct only selected points in the scene, usually by extracting and matching features. Dense methods [18; 33; 7; 52] maintain a dense scene representation which is used for reconstruction and tracking, generally based on photometric loss.

The success of radiance fields in novel view synthesis has made them a popular scene representation for dense SLAM. The first SLAM system using a radiance field as the only scene representation was iMAP [48], using a multi-layer perceptron (MLP) as its scene representation. NICE-SLAM [68] used an explicit hierarchical feature grid along with pre-trained MLPs to represent the scene, allowing it to better represent large-scale scenes. NICE-SLAM [68] was extended to only require RGB data instead of RGB-D in NICER-SLAM [69] by making use of monocular depth and normal estimators and optical flow loss. GO-SLAM [66] introduced global Bundle Adjustment (BA) and loop closure for an implicit SLAM system using an MLP representing a signed-distance field, greatly increasing tracking performance compared to previous radiance field-based SLAM methods. SLAM systems using 3D Gaussian Splatting (3DGS) [21] as their scene representation have also recently been developed [20; 59; 30; 16]. By using 3DGS these systems can achieve higher fidelity scene representations without sacrificing speed. We use MonoGS [30] as the backbone of our active SLAM system.

**Active SLAM**    In the active SLAM task, the agent must construct a trajectory to explore and map the environment. In order to create an accurate map there should be a trade-off between exploring new regions and reducing the uncertainty of the map and pose estimates [22; 40; 3].

Active SLAM systems are typically divided into three components [27; 37] – candidate goal identification, utility computation, and action planning and execution. Frontier-based exploration (FBE) [58] is a widely used technique for proposing candidate goals [47; 2; 53; 49]. The utility calculated in the second stage of active SLAM algorithms generally seeks to capture uncertainty [27; 37] – specific utility functions are often drawn from either Information Theory (IT) [43] or the Theory of Optimal Experimental Design (TOED) [35]. Metrics from IT are based on entropy, whereas those from TOED are based on covariance. Many TOED-based methods formulate objectives using the Fisher Information Matrix (FIM) because its inverse is the Cramer-Rao lower bound of the covariance matrix and the FIM is generally sparser than the covariance [37]. This approach is often taken by filter-based SLAM algorithms [10; 55; 22], as the filter produces a covariance of the state estimate. For the action planning and execution, various path planning algorithms are used, including RRT [24] in Vallvé & Andrade-Cetto [54]; Huang & Gupta [17] and A* [13] in Kim & Eustice [22]. We follow

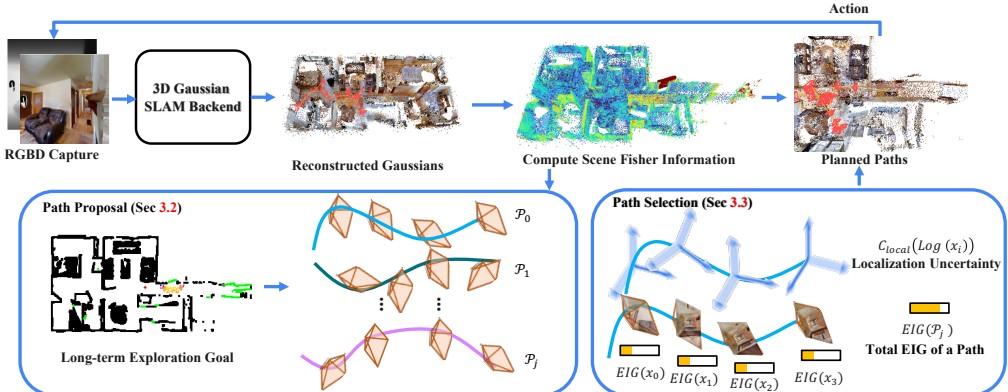

Figure 1: **An Illustration of Our Active SLAM System** The Active Gaussian SLAM system proposes paths based on the Fisher Information about the parameters of the 3D Gaussian parameters. The best path and action along the path is selected with respect to both the information gain and localization accuracy. The Active SLAM system progressively improves the mapping and localization during exploration

this three-component approach to active SLAM, using standard techniques for the first and third components (FBE and A*, respectively) and a novel utility formula based on Fisher Information.

**Uncertainty quantification for radiance fields** The vast majority of previous work on uncertainty quantification for radiance fields has been for post-processing scenes [44; 45; 50; 12], view selection [34; 19; 50; 25] or active view selection [34; 50; 19], all of which assume the input images are posed. Active neural mapping [61] uses neural variability, that is the prediction robustness against random weight perturbations, as an estimate of uncertainty to actively map a scene with ground truth poses provided. Fisher-RF [19] also performs active scene mapping with ground truth poses provided, based on an approximation of the Fisher Information of views along candidate paths. Zhan et al. [64] perform active reconstruction without ground truth camera poses, however unlike us they only evaluate on small scenes and limit the camera trajectories to be inwards facing. Unlike our method, these prior works only model scene uncertainty, not localization uncertainty which is a key consideration for active SLAM.

## 3 METHOD

In this section, we first introduce the preliminary background of 3D Gaussian Splatting SLAM [30] and FisherRF [19], which are the foundation for our algorithm in Section 3.1. We then discuss our information-driven path proposal algorithm in Sec. 3.2 and our path selection algorithm in Sec. 3.3.

### 3.1 PRELIMINARY

In 3D Gaussian Splatting (3DGS) [21], the scene is represented by a set of 3D Gaussians whose color and opacity are learned via a rendering loss. The rendered image depends on the 2D Gaussians $\mathcal{N}(\boldsymbol{\mu}_I, \boldsymbol{\Sigma}_I)$ that are projections $\pi(.)$ of 3D Gaussians $\mathcal{N}(\boldsymbol{\mu}_W, \boldsymbol{\Sigma}_W)$ in world coordinates. The projection of a 3D Gaussian in the world frame to a 2D Gaussian on the image plane can be written as

$$\boldsymbol{\mu}_I = \pi(\mathbf{x}^{-1} \cdot \boldsymbol{\mu}_W) \, , \boldsymbol{\Sigma}_I = \mathbf{J}\mathbf{W}\boldsymbol{\Sigma}_W\mathbf{W}^T\mathbf{J}^T \, , \tag{1}$$

where $\mathbf{x} \in \boldsymbol{SE}(3)$ is the world-to-camera transformation, , and $\mathbf{J}$ is the Jacobian of the linear approximation of the projective transformation and $\mathbf{W}$ is the rotational component of $\mathbf{x}$. Each pixel's color $C_p$ is then calculated from the 2D Gaussians using $\alpha$-blending for the $N$ ordered points on the 2D splat that overlaps the pixel. That is,

$$C_p = \sum_{i=1}^{N} \mathbf{c}_i \alpha_i \prod_{j=1}^{i-1} (1 - \alpha_j), \tag{2}$$

where $\mathbf{c}_i$ is the learned color associated with each Gaussian, and $\alpha_i$ is calculated using the covariance of the corresponding 2D Gaussian following Yifan et al. [62] then multiplying with a learned per-point opacity. Following MonoGS [30], we omit the view-dependence of the color as it greatly saves the number of parameters and memory usage in large-scale scenes while maintaining satisfactory performance. The rendering at a given camera location $\mathbf{x}$ with 3DGS parameters $\mathbf{w}$ can be considered as a function $f(\mathbf{x}, \mathbf{w})$.

MonoGS [30] found the Jacobian of the current camera pose $\mathbf{x}$ with respect to the parameters of 3D Gaussians $\boldsymbol{\mu}_I$ and $\boldsymbol{\Sigma}_I$ using the chain rule:

$$\frac{\partial \boldsymbol{\mu}_I}{\partial \mathbf{x}} = \frac{\partial \boldsymbol{\mu}_I}{\partial \boldsymbol{\mu}_C} \frac{\mathcal{D}\boldsymbol{\mu}_C}{\mathcal{D}\mathbf{x}} \ , \tag{3}$$

$$\frac{\partial \boldsymbol{\Sigma}_I}{\partial \mathbf{x}} = \frac{\partial \boldsymbol{\Sigma}_I}{\partial \mathbf{J}} \frac{\partial \mathbf{J}}{\partial \boldsymbol{\mu}_C} \frac{\mathcal{D}\boldsymbol{\mu}_C}{\mathcal{D}\mathbf{x}} + \frac{\partial \boldsymbol{\Sigma}_I}{\partial \mathbf{W}} \frac{\mathcal{D}\mathbf{W}}{\mathcal{D}\mathbf{x}} \ , \tag{4}$$

Let $\tau \in \mathfrak{se}(3)$ and define the (left) partial derivative on the manifold as:

$$\frac{\mathcal{D}f(\mathbf{x})}{\mathcal{D}\mathbf{x}} \triangleq \lim_{\tau \to 0} \frac{\mathrm{Log}(g(\mathrm{Exp}(\tau) \circ \mathbf{x}) \circ g(\mathbf{x})^{-1})}{\tau} \ , \tag{5}$$

where $\circ$ is a group composition, and Exp and Log are the exponential and logarithmic mappings between Lie algebra and Lie Group.

Fisher Information is a measurement of the information that a random variable $\mathbf{y}$ carries about an unknown parameter $\mathbf{w}$ of a distribution that models $\mathbf{y}$. In the problem of novel view synthesis, we are interested in measuring the observed information of a radiance field with parameters $\mathbf{w}$ at a camera pose $\mathbf{x}$ using the negative log-likelihood of the image observation $\mathbf{y}$ taken from that pose:

$$-\log p(\mathbf{y}|\mathbf{x}; \mathbf{w}) = (\mathbf{y} - f(\mathbf{x}, \mathbf{w}))^T (\mathbf{y} - f(\mathbf{x}, \mathbf{w})), \tag{6}$$

where $f(\mathbf{x}, \mathbf{w})$ is the rendering model. Under regularity conditions [41], the Fisher Information of $-\log p(\mathbf{y}|\mathbf{x}; \mathbf{w})$ is the Hessian of Eq. 6 with respect to $\mathbf{w}$, denoted $\mathbf{H}''[\mathbf{y}|\mathbf{x}, \mathbf{w}]$.

FisherRF [19] addressed the active view selection problem that starts with a training set of views $D^{train}$ and aims to select the next best view from a set of candidate $SE(3)$ camera poses $\mathbf{x}_i^{acq} \in D^{pool}$ without obtaining the image $\mathbf{y}_i^{acq}$ at the camera pose $\mathbf{x}_i^{acq}$. The next best view is chosen by finding:

$$\arg\max_{\mathbf{x}_i^{acq} \in D^{pool}} \mathrm{tr}\left(\mathbf{H}''[\mathbf{y}_i^{acq}|\mathbf{x}_i^{acq}, \mathbf{w}] \, \mathbf{H}''[\mathbf{w}|D^{train}]^{-1}\right), \tag{7}$$

where $\mathbf{w}$ is the initial estimate of model parameters using current training set $D^{train}$. $\mathbf{H}''[\mathbf{w}|D^{train}]^{-1}$ can be computed by summing the Hessians of model parameters across all different views in $\{D_{train}\}$ before inverting. The key of this algorithm is that the Fisher Information $\mathbf{H}''[\mathbf{y}_i^{acq}|\mathbf{x}_i^{acq}, \mathbf{w}]$ does not depend on the label $\mathbf{y}_i^{acq}$ of the acquisition sample $\mathbf{x}_i^{acq}$. Therefore, it is feasible to compute the Expected Information Gain (EIG) before visiting the potential view candidate $\mathbf{x}_i^{acq}$. However, the number of optimizable parameters is typically more than 20 million, which means it is impossible to compute without sparsification or approximation. In practice, FisherRF [19] applies a Laplace approximation [6; 28] that approximates the Hessian matrix with its diagonal values plus a log-prior regularizer $\lambda I$

$$\mathbf{H}''[\mathbf{y}|\mathbf{x}, \mathbf{w}] \simeq \mathrm{diag}(\nabla_{\mathbf{w}} f(\mathbf{x}, \mathbf{w})^T \nabla_{\mathbf{w}} f(\mathbf{x}, \mathbf{w})) + \lambda I. \tag{8}$$

To understand the usage of EIG, In Fig. 2, we plot the Peak-signal-to-noise ratio (PSNR) vs EIG at sampled poses and show some example renderings to give a sense of how the EIG is related to the PSNR and the rendering quality. More importantly, unlike PSNR, the EIG can be computed without ground truth images, making it possible to perform view selection during exploration.

## 3.2 Information-driven Path Proposal

We consider an agent moving in a 2D plane (e.g. a wheeled robot), so we construct a 2D occupancy map for path planning by ground projecting the means of 3D Gaussians.

Frontier-based exploration [58] is incorporated into our algorithm to provide candidates for view selection. The frontiers are defined to be points on the boundary between free space and unobserved

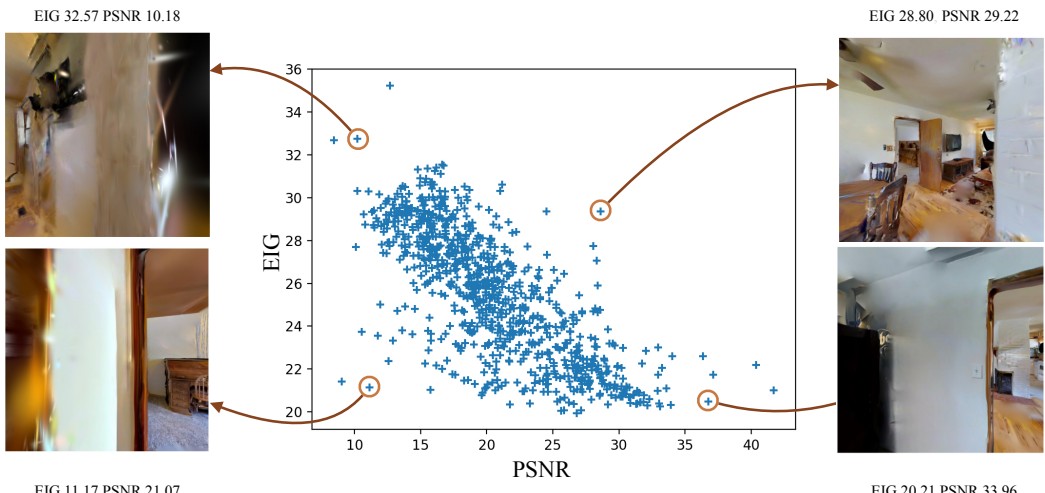

EIG 32.57 PSNR 10.18

EIG 28.80  PSNR 29.22

EIG 11.17 PSNR 21.07

EIG 20.21 PSNR 33.96

Figure 2: **Scatter Plot of EIG vs. PSNR** We plot the EIG and PSNR at sampled poses in the Cantwell scene of the Gibson dataset. The figure corroborates the intuition that the robot expects to gain little information (low EIG) at well reconstructed region (high PSNR) and gain much information (high EIG) at a poorly reconstructed region (low PSNR).

space (not marked as free or occupied) in our occupancy map. We then form an initial set of candidate poses $\mathcal{T}_I$ by sampling poses around each point in the largest frontier. If there are no frontiers, we instead sample poses around the center of each Gaussian to form $\mathcal{T}_I$. We then evaluate the Expected Information Gain (EIG) for each pose $\mathbf{x}_i^{acq} \in \mathcal{T}_I$, given by

$$\text{EIG}(\mathbf{x}_i^{acq}) = \text{tr}\left(\mathbf{H}''[\mathbf{y}_i^{acq}|\mathbf{x}_i^{acq}, \mathbf{w}] \, \mathcal{I}(\mathbf{w})^{-1}\right), \tag{9}$$

as a preliminary selection metric to form our final candidate target poses set $\mathcal{T}_F$. Thus, we can identify multiple coarse directions for explorations where we can propose multiple paths for detailed path planning and path selection. FisherRF [19] uses $\mathbf{H}''[\mathbf{w}|D^{train}]$ as an approximation for the observed Fisher Information $\mathcal{I}(\mathbf{w})$ by computing the Hessians on the training set. This is also known as empirical Fisher Information, whose limitations have been widely discussed by Kunstner *et al.* [23] and Marten *et al.* [29]. In most scenarios, this is a reluctant design choice because the distribution of $\mathbf{x} \sim p(\mathbf{x})$ is unknown (i.e., the distribution of all possible images). However, $\mathbf{x} \sim p(\mathbf{x})$ in our case is tractable because it represents the possible locations where we can take an observation for the environment, a.k.a. the free space of our map. Therefore, unlike FisherRF [19] as described in Eq. 7, we propose to use Monte-Carlo sampling to compute the Fisher Information of the current model

$$\mathcal{I}(\mathbf{w}) = \mathbb{E}_{\mathbf{x} \sim p(\mathbf{x})}\left[\mathbf{H}''[\mathbf{y}|\mathbf{x}, \mathbf{w}]\right] \simeq \sum_{k=1}^{N} \mathbf{H}''[\mathbf{y}_k|\mathbf{x}_k, \mathbf{w}], \quad \mathbf{x}_k \sim p(\mathbf{x}) \tag{10}$$

where $p(\mathbf{x}_k)$ is approximated with a uniform distribution of camera poses in the free space of the current map. Besides, we also uniformly initialize 3D Gaussians in the space, which will be subsequently updated with rendering losses for visited regions. In Fig. 2, we show the relationship between PSNR and EIG on sampled poses in the Cantwell scene. The result aligns with common sense that EIG should decrease as PSNR increases. We also show some cases on the lower-left and upper-right of the scatter plot. For the lower left capture, even though the left part is poorly reconstructed, most of view is occupied by textureless wall, leading to low EIG score. For the upper right capture, even though the scene has a moderate reconstruction, the content is rich so our algorithm returns a high EIG score.

Finally, we compute paths towards each pose in $\mathcal{T}_F$ with the A* algorithm [13] using the occupancy map, selecting which path to follow as described in Section 3.3. The path can be defined as an ordered set of camera poses from the current location $x_t$ at exploration step $t$ to the frontier points $x_T^j$.

$$\mathcal{P}_j = \{x_{t+1}^j, \ldots, x_T^j\} \tag{11}$$

## 3.3 PATH SELECTION WITH LOCALIZATION UNCERTAINTY

Following FisherRF [19], the EIG for 3D Gaussian parameters along a path $\mathcal{P}_j$ can be computed as:

$$\sum_{\mathbf{x}_i \in \mathcal{P}_j} \text{EIG}_{\mathcal{P}_j,i}(\mathbf{x}_i), \quad \text{EIG}_{\mathcal{P}_j,i}(\mathbf{x}_i) = \text{tr}\left(\mathbf{H}''[\mathbf{y}_i|\mathbf{x}_i,\mathbf{w}] \,\mathcal{I}_{\mathcal{P}_j,i}(\mathbf{w})^{-1}\right) \tag{12}$$

where $\mathcal{I}_{\mathcal{P}_j,i}(\mathbf{w})$ takes the mutual information along the path into account as follows

$$\mathcal{I}_{\mathcal{P}_j,i}(\mathbf{w}) = \mathbf{H}''[\mathbf{w}] + \sum_{\mathbf{x}_t \in \mathcal{P}_j, t<i} \mathbf{H}''[\mathbf{w}|\mathbf{x}_t]. \tag{13}$$

where $\mathbf{H}''[\mathbf{x}]$ is short for $-\nabla^2 \log p(w)$ for clarity. If solely maximizing the EIG, the robot will be more likely to explore unvisited regions. However, exploring regions that have not been well reconstructed also means the agent would have the risk of worse localization accuracy due to noise and ambiguities in the unreconstructed regions during pose optimization. The cost of localization must thus be considered during path planning to balance the importance of exploring new environments with maintaining localization accuracy. Please note it is possible to add a weighted factor for the EIG. This is omitted in our experiments because we cap the maximum steps, as the EIG is used to choose short-term exploration goals, so the proposed paths have similar lengths. We propose to use Fisher Information as a measurement for the localization uncertainty that is also necessary for effective path planning for active SLAM algorithms. During optimization, we essentially optimize on the logarithmic mapping of $\tau_i \triangleq \text{Log}(\mathbf{x}_i)$ of our camera pose. By the Cramér–Rao bound, the covariance of $\tau_i \in \mathfrak{se}(3)$ can be lower-bounded with the inverse of Fisher Information matrix $\mathcal{I}(\tau_i)$:

$$\text{Cov}(T(\hat{\tau_i})) \geq \mathcal{I}(\tau_i)^{-1} \tag{14}$$

where $T(\tau_i)$ is an unbiased estimator for $\tau$ solved by iteratively optimizing photo-metric loss. Hence, we can define the localization cost $C_{local}$ at a pose $\mathbf{x}_i$ in terms of $\tau_i$ as:

$$C_{local}(\tau_i) = \log \det(\nabla_{\tau_i} f(\tau_i, \mathbf{w})^T \nabla_{\tau_i} f(\tau_i, \mathbf{w})) \tag{15}$$

Matsuki *et al.* [30] computed the Jacobians of camera pose with respect to the mean and covariances of each gaussian $\frac{\partial \boldsymbol{\mu}_I}{\partial \mathbf{x}}$ and $\frac{\partial \boldsymbol{\Sigma}_I}{\partial \mathbf{x}}$. However, we need to compute the Jacobian of $\tau_i$ with respect to the rendering output:

$$\nabla_{\tau_i} f(\tau_i, \mathbf{w}) = \frac{\partial f(\tau_i, \mathbf{w})}{\partial \tau_i} = \begin{bmatrix} \frac{\partial f(\tau_i,\mathbf{w})}{\partial \boldsymbol{\mu}_I} & \frac{\partial f(\tau_i,\mathbf{w})}{\partial \boldsymbol{\Sigma}_I} \end{bmatrix} \begin{bmatrix} \frac{\mathcal{D}\boldsymbol{\mu}_C}{\mathcal{D}\tau_i} \\ \frac{\mathcal{D}\mathbf{W}}{\mathcal{D}\tau_i} \end{bmatrix} \tag{16}$$

Without loss of generality, the path of exploration can be selected by minimizing the total cost for all viewpoints $\mathbf{x}_i$ along a path $\mathcal{P}_j$:

$$\arg\min_{\mathcal{P}_j} \sum_{\mathbf{x}_i \in \mathcal{P}_j} C_{local}(\text{Log}(\mathbf{x}_i)) - \eta \log(\text{EIG}_{\mathcal{P}_j,i}(\mathbf{x}_i)) \tag{17}$$

where $\eta$ is a hyper-parameter controlling the importance between EIG and localization accuracy. The agent can then explore the environment with planned path $\mathcal{P}$. Our active SLAM system constantly updates the map, and we replan using our active path planning algorithm if we detect the agent is getting close to a possible obstacle or upon reaching the end of the previously selected path.

## 4 EXPERIMENTS

### 4.1 EXPERIMENTAL SET-UP

**Dataset** Our algorithm is evaluated in the Habitat Simulator [51] on the Gibson [57] and Habitat-Matterport 3D (HM3D) [39] datasets, which are comprised of indoor scenes reconstructed from scans of real houses. For Gibson we use all the scenes in the val split. For HM3D we use 5 scenes

from the train split. We adopt the default start point in the Habitat Simulator as the starting point for exploration in each scene. The total number of steps for each experiment is 2000. The system takes color and depth images at the resolution of 800x800 and outputs a discrete action at each step. The action space consists of MOVE FORWARD by 5cm, TURN LEFT, and TURN RIGHT by $5°$. The field of view (FOV) is set to $90°$ vertically and horizontally. Please refer to the appendix for more details about the evaluation split and other hyper-parameters.

**Metrics** We evaluated our method using the Peak-signal-to-noise ratio (PSNR), Structural Similarity Index Measure SSIM [56], Learned Perceptual Image Patch Similarity (LPIPS) [65] for RGB rendering and mean absolute error (MAE) for depth rendering as a metric for scene reconstruction quality. We calculate these metrics using 2000 points uniformly sampled from the movement plane of the agent in the scene, discarding any points that are not navigable. We argue the rendering quality reflects both reconstruction quality and the pose accuracy because high tracking accuracy would help the training of the 3D Gaussian Splatting model. Meanwhile, misaligned pose accuracy will lead to misaligned rendering at test time thus leading to inferior results. Following previous approaches [11; 4], We also use coverage in $m^2$ and $\%$ as evaluation metrics. To evaluate the pose estimation accuracy we use the root mean squared average tracking error (RMSE ATE). Please note that for active SLAM the trajectories for each method are different so the RMSE ATE should only be considered along with other metrics such as coverage.

**Baselines** We compare to two exploration methods which assume ground truth pose: UPEN [11] and Active Neural Mapping (active-INR) [61]. UPEN uses an ensemble of models to predict and estimate the epistemic uncertainty of the occupancy map outside the field of view and leverages these to construct paths that reduce the uncertainty of the occupancy prediction. We also report the results of UPEN(gt), which uses ground truth pose as a reference because we find UPEN failed on some scenes due to localization failure. Active-INR aims to minimize the neural variability, that is the prediction robustness against random weight perturbation, of its signed distance field scene representation. We also compare our method with Active Neural SLAM (ANS) [4], explORB [36] and Frontier Based Exploration (FBE) [58] without ground truth pose provided. ANS learns to predict a map and pose estimate and a global goal based on them, which is then reached using a combination of a classical path planner and a trained local policy. ExplORB [36] adds possible loop closure based on the co-visibility of future pose and existing poses based on existing landmarks. It computes the Fisher Information of the Hessian on the pose graph optimization. FBE can be considered an ablation of our method to validate the importance of considering the scene and localization uncertainty, where instead of choosing paths using Eq. 17 we instead select the frontier based on the ratio of its area to the agent's distance from it.

To make a fair comparison of the rendering quality, we run all the baselines using the MonoGS [30] backend for reconstruction. We run UPEN and FBE online but for ANS, active-INR and ExplORB we record and play back trajectories obtained using their codebase. Because the forward step size for ANS is much larger than for our method, we interpolate the trajectory so that the forward step size matches that of our method to make the steps comparable. For ExplORB, since the official implementation is based on MoveBase, which uses velocity commands, we sample the trajectory at 5 Hz. This will limit the distance change between frames to around 4cm and $5°$ for the linear and angular distance, respectively, with the maximal linear and angular velocities as 20 cm/s and 0.5 rad/s, respectively. We also found that ANS and active-INR failed on some scenes due to localization failure. ANS produces a pose estimate (using information from noisy pose sensors not provided to our pipeline), so we set the pose estimate of the MonoGS backend to the one from ANS. As active-INR does not produce a pose estimate we evaluate it using the ground-truth pose.

## 4.2 COMPARISON AGAINST PREVIOUS METHODS

Table 1 shows the results of our method AG-SLAM and the baselines for exploration in scenes from the Gibson dataset, and Table 2 shows the results of our method and some baselines on HM3D. Note that the percentage coverage reported is the average percentage coverage per scene. As the scenes are different sizes this means that a method can have a lower percentage coverage with a higher area coverage, for example if it has better coverage in larger scenes but worse in smaller scenes. Our AG-SLAM outperforms the baselines on all metrics.

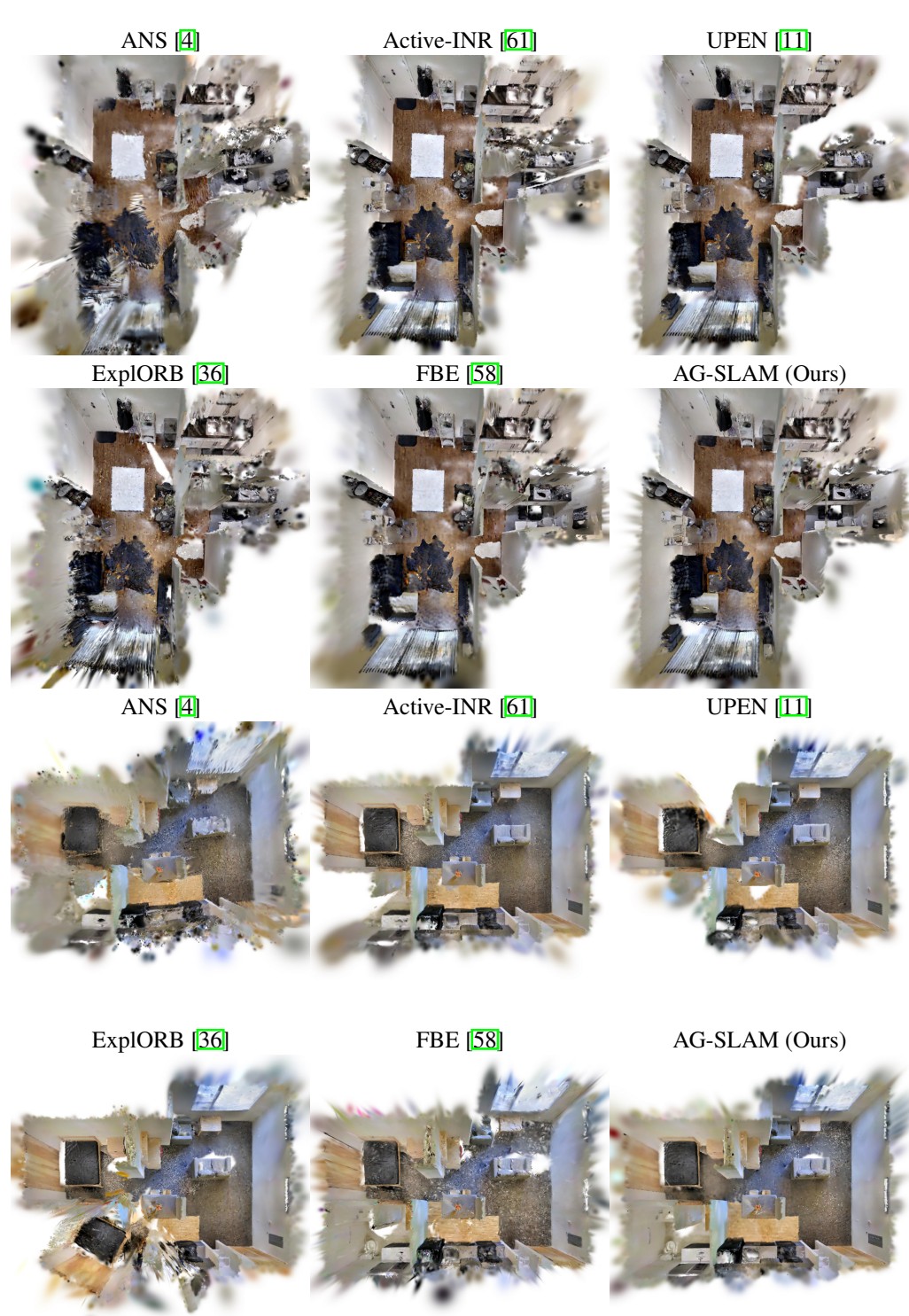

Figure 3: **Qualitative Comparison for Final Scene Reconstruction on Gibson Dataset** Greigsville (top) and Ribera (bottom) scenes. We provide top-down rendering for different methods. Note that UPEN and Active-INR use GT pose in this visualization.

UPEN [11]  FBE [58]  AG-SLAM (Ours)

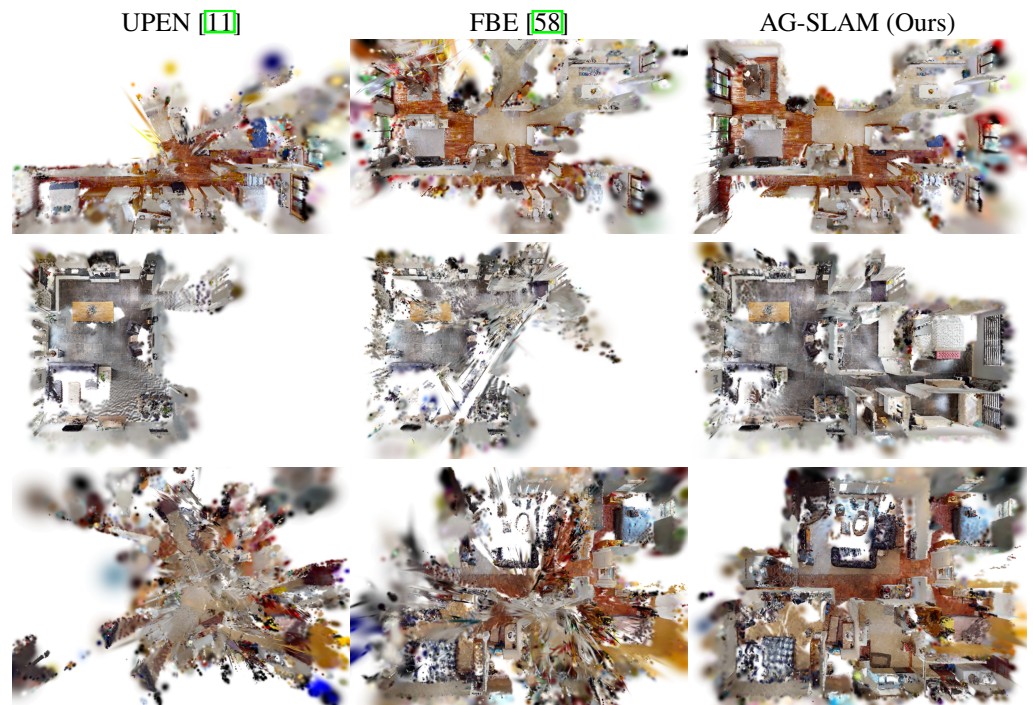

Figure 4: **Qualitative Comparison for Final Scene Reconstruction on Habitat-Matterport 3D Dataset** mscxX4KEBcB (top), oPj9qMxrDEa (middle) and QKGMrurUVbk (bottom) scenes. We provide top-down rendering for different methods.

Table 1: **Evaluation of AG-SLAM** and baselines on scenes from the Gibson dataset

| Method | PSNR ↑ | SSIM ↑ | LPIPS ↓ | Depth MAE ↓ | RMSE ATE ↓ | Coverage ($m^2$) ↑ | Coverage (%) ↑ |
|---|---|---|---|---|---|---|---|
| ANS | 16.34 | 0.6818 | 0.3923 | 0.3886 | 0.1105 | 10.49 | 90.50 |
| Active-INR (gt) | 22.66 | 0.7652 | 0.2164 | 0.1528 | - | 9.20 | 78.34 |
| UPEN (gt) | 21.31 | 0.7325 | 0.2714 | 0.1696 | - | 8.79 | 76.17 |
| UPEN | 16.44 | 0.6678 | 0.4134 | 0.4841 | 0.5158 | 8.58 | 75.82 |
| ExplORB | 18.99 | 0.7175 | 0.3994 | 0.2664 | 0.2296 | 9.00 | 76.83 |
| FBE | 21.45 | 0.7618 | 0.2126 | 0.1028 | 0.0168 | 10.64 | 85.81 |
| AG-SLAM (ours) | **23.08** | **0.7959** | **0.1794** | **0.0763** | **0.0155** | **11.23** | **91.30** |

Table 2: **Evaluation of AG-SLAM** and baselines on scenes from the HM3D dataset

| Method | PSNR ↑ | SSIM ↑ | LPIPS ↓ | Depth MAE ↓ | RMSE ATE ↓ | Coverage (m²) ↑ | Coverage (%) ↑ |
|---|---|---|---|---|---|---|---|
| UPEN (gt) | 15.58 | 0.5175 | 0.3936 | 0.4548 | - | 12.69 | 53.78 |
| UPEN | 12.23 | 0.4795 | 0.5157 | 0.7356 | 0.4393 | 10.72 | 44.96 |
| FBE | 15.80 | 0.5952 | 0.4392 | 0.4085 | 1.2004 | 15.69 | 66.44 |
| AG-SLAM (ours) | 18.74 | **0.6277** | **0.3534** | **0.1757** | **0.0208** | **19.70** | **80.96** |

We further qualitatively compare the reconstruction qualities after active exploration in Fig. 3 and Fig. 4, and the trajectories in Fig. 5. We can see that AG-SLAM does not have major errors from failed localization and we have fewer gaps in the scenes than other methods. For example, in the Ribera scene all methods except for us and FBE miss the bathroom in the bottom left, and FBE misses more of the area around the sofa than us. For the trajectories, we show the estimated and ground truth trajectories for the Cantwell scene from the Gibson dataset. Cantwell is a relatively large and challenging scene, so it is suitable for showing the differences between methods. We show only a few baselines to keep the figure legible. We can see that Active-INR stays in a smaller area than the rest of the methods. ANS often goes close to walls, whereas FBE and AG-SLAM are generally more

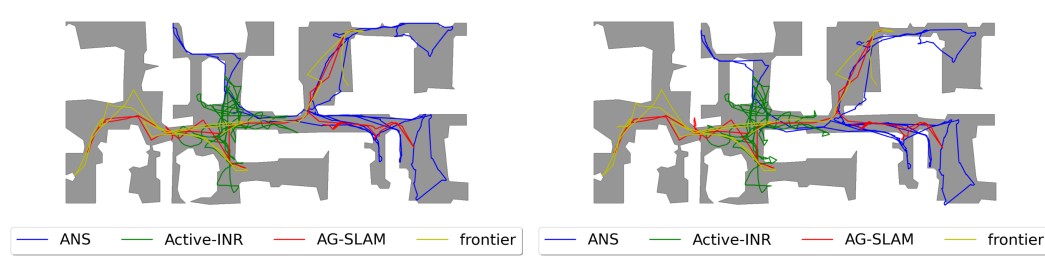

Ground truth                                           Estimated

Figure 5: **Qualitative Comparison for Trajectories** on Cantwell from Gibson Dataset. Left is the ground truth trajectory, right is the estimated trajectory.

towards the center of corridors or rooms. ANS also extends further into rooms than these methods. FBE does not go to the region on the bottom right and ANS does not go to the region in the bottom left, whereas our AG-SLAM efficiently visits the main areas of the scene.

### 4.3 ABLATION STUDY

To validate the effectiveness of using localization uncertainties, we performed an ablative study of our SLAM system with and without localization uncertainty for the Gibson dataset in Table. 3. As can be seen, the average trajectory error is also much lower with the localization term than without. In addition, the model performs consistently better for rendering when considering the localization uncertainty, as localization accuracy plays a key role in the optimization of Gaussian Splatting representation, and the quality of rendering is also dependent on localization accuracy. The percentage coverage is slightly better when not including the localization term; this can be explained by the method focusing more on exploration when it no longer balances it with localization accuracy. The area coverage is higher with the localization term, however, indicating that better localization could benefit coverage in larger scale scenes.

Table 3: **Ablation Study of Localization Uncertainty Term on Scenes from the Gibson Dataset.** We compare our method with and without the localization uncertainty term to validate that including it provides improvements on both localization and reconstruction

| Method | PSNR ↑ | SSIM ↑ | LPIPS ↓ | Depth MAE ↓ | RMSE ATE ↓ | Coverage $(m^2)$ ↑ | Coverage (%) ↑ |
|---|---|---|---|---|---|---|---|
| w.o. Localization Uncertainty | 22.35 | 0.7830 | 0.3089 | 0.0823 | 0.1890 | 10.85 | **91.75** |
| AG-SLAM | **23.08** | **0.7959** | **0.1794** | **0.0763** | **0.0155** | **11.23** | 91.30 |

## 5 CONCLUSION

Recent SLAM methods employ the 3D Gaussian Splatting (3DGS) representation of the world, enabling a volumetric rendering as a measurement prediction. In this paper, we introduced active pose selection for 3DGS-based SLAM. Our AG-SLAM balances the information gain with respect to both location and the map. We mathematically formulated the expected information gain using the Fisher Information matrix and the Cramer-Rao Lower Bound. We evaluate our method for active SLAM on scenes from the Gibson [57] and Habitat-Matterport 3D [39] datasets, in terms of the rendering quality, coverage and average tracking error. We show our uncertainty-based criteria for path selection improves over using frontier-based exploration [58] with a selection criteria that uses the frontier area and distance. We also compare our method with four recent state-of-the-art methods and show that AG-SLAM has superior performance.

To enable AG-SLAM to support more robotics applications, future work could extend AG-SLAM to consider movement with higher degrees of freedom (DOF) than the currently supported 3DOF. Incorporating semantic features [67; 46; 38] to allow for grounding language to the scene would also enable many robotics and computer vision applications.

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
