# OpenReview forum: "AG-SLAM: Active Gaussian Splatting SLAM"
_ICLR.cc/2025/Conference — ICLR 2025 Conference Withdrawn Submission_

### Official Review · Reviewer_REtt · 2024-11-03

**Soundness:** 3
**Presentation:** 3
**Contribution:** 2
**Rating:** 6
**Confidence:** 4

**Summary:**

The paper presents Active Gaussian Splatting SLAM, an active SLAM pipeline which builds on top of MonoGS and uses 3DGS as its underlying 3D representation. Unlike typical SLAM, active SLAM incorporates planning to maximize exploration efficiency. It leverages Fisher information to assess map and localization uncertainty, which are jointly considered in the active planning process to enable exploration while minimizing localization failures.

**Strengths:**

To my knowledge, this is one of the first works to apply active SLAM to 3D Gaussian Splatting. The proposed method is sound and the motivation for jointly modeling mapping and localization uncertainty is clear.
The evaluation supports the effectiveness of the approach

**Weaknesses:**

It is unclear how much overhead there is to compute the fisher information. Eq. 16 requires the Jacobian wrt to the rendering output, does this mean that the Jacobian is (num_pixels)x6?

Though it's a SLAM system, run time is not reported which makes it hard to assess whether the approach is practical.

The PSNR evaluation is somewhat unfair. MonoGS directly minimizes PSNR, whereas other SLAM methods optimize different objectives. For a fairer comparison, all methods should perform pose refinement using MonoGS, with the SLAM trajectory as initialization.

**Questions:**

How is the occupancy map size determined? One advantage of 3DGS over NeRF is that it doesn’t require a bounding box. Was this necessary to make Equation 10 tractable? Clarifying any potential downsides to this choice would be beneficial.

I’m also surprised that ExplORB has a worse Absolute Trajectory Error (ATE) than AG-SLAM, which uses MonoGS as its backend, especially if ExplORB has loop closure enabled. Is there an explanation for this? (In MonoGS paper, ORB-SLAM outperforms it on the TUM RGB-D dataset.)

Do you incorporate kinematic information into the pose estimation?

---

### Official Review · Reviewer_6TTw · 2024-11-03

**Soundness:** 1
**Presentation:** 2
**Contribution:** 2
**Rating:** 3
**Confidence:** 4

**Summary:**

This paper proposes an approach for active SLAM using a 3D Gaussian splatting map representation by considering localization and reconstruction uncertainty. It extends 3DGS SLAM with frontier-based exploration [58] and the expected information gain based exploration approach in FisherRF [19] by considering the uncertainty of the 3D Gaussian reconstruction parameters. Additionally, the approach proposes to include a term which measures uncertainty through the fisher information matrix for the camera pose parameters. Evaluation is conducted in embodied AI simulations Habitat using the Habitat-Matterport 3D dataset and Gibson and improvements over previous methods is demonstrated in terms of reconstruction quality and localization accuracy.

**Strengths:**

- The idea of combining reconstruction and localization accuracy in the context of SLAM with 3D Gaussian splatting seems novel and interesting.
- The experiments indicate improvements over previous methods and an ablation in reconstruction quality and localization accuracy in several simulated sequences.

**Weaknesses:**

- The method description needs significant improvements for clarity of notation and explanations. For instance, it lacks proper introduction and definition of the expected information gain that is optimized.
- The proposed calculation of I(w) does not make sense as it should measure the entropy of the 3DGS map parameters after initialization with the previous training views before integrating further view information. It is not clear why sampling other views than the training views should improve information about this initial parameter estimate.
-  It is not clear what the quantity in eq. 13 is and why it should be considered. For a mutual information, the conditional entropy should be subtracted. It seems what the authors would like to achieve is to compute the expected information gain along a path. For this a simulation of the measurements along the path and incremental integration of measurements into the 3DGS would be required though to be able to measure the conditional entropy given the sequence of measurements.
- Representing poses as Lie algebra elements at identity (i.e., axis-angle vectors for the rotation part) is generally suboptimal for pose optimization and covariance representation as the rotation representation is not unique and has singularities at multiples of 2\pi.
- l. 503, what quantifies a "main area of the scene" ?
- Restricting the trajectory length to 2000 steps seems artificial. Rather, a significantly larger number steps should be used as time out and the approaches should be given chance to explore the whole environment. Which method explores the scene faster? Is there a trade off between exploration time, localization and reconstruction accuracy ? Do all methods explore the environment fully?
- How are hyperparameters of the proposed approach and the baselines obtained ? Is the comparison fair?
- Why are the specific HM3D scenes chosen for evaluation? Have parameters of the methods been tuned on different scenes?

Minor comments:
- "so we set the pose estimate of the MonoGS backend to the one from ANS" - this should be phrased the other way round.
- please use booktabs standard style for format tables.

**Questions:**

- See questions in paper weaknesses. The most important concerns are about the soundness of the methodology and the experiment setup (hyperparameter tuning, limitation to 2000 steps).

---

### Official Review · Reviewer_oh6K · 2024-11-04

**Soundness:** 2
**Presentation:** 2
**Contribution:** 2
**Rating:** 5
**Confidence:** 3

**Summary:**

The paper presents AG-SLAM, an active SLAM system that leverages 3D Gaussian Splatting (3DGS) to enhance scene reconstruction.
While traditional SLAM systems prioritize accurate localization, radiance field-based methods emphasize high-quality scene reconstruction, often neglecting self-localization accuracy. The proposed approach addresses both issues simultaneously, enhancing both localization precision and reconstruction quality.

AG-SLAM focuses on balancing exploration with localization uncertainty by integrating Fisher Information into its framework. This dual-objective strategy, as argued in the paper, maximizes information gain for efficient mapping while reducing localization errors.

Experimental results on the Gibson and Habitat-Matterport 3D datasets demonstrate AG-SLAM’s good performance in both scene reconstruction quality and trajectory estimation accuracy compared to other active SLAM methods.

**Strengths:**

1. **Interesting Combination**: AG-SLAM proposes a system that integrates 3D Gaussian Splatting with active SLAM, setting an interesting direction for online scene reconstruction. This approach extends the research of 3DGS SLAM to the active SLAM problem setting.

2. **Balanced Objective Function**: The paper introduces an effective objective function that manages the trade-off between exploration and reducing the uncertainty of the estimated agent’s pose and environment map quality.

3. **Performance**: AG-SLAM consistently outperforms competing methods across metrics such as PSNR, SSIM, and LPIPS, indicating its effectiveness in both RGB and depth rendering. The experiments provide evidence of AG-SLAM's benefits in terms of scene coverage and localization accuracy.

**Weaknesses:**

**1. Limited Contribution**: While I appreciate the author’s efforts in integrating 3D Gaussian Splatting (3DGS) with active SLAM, I think the technical contribution presented in this paper to be somewhat limited and may not be insufficient for publication at the ICLR level. The core methodology lacks substantial innovation, as the primary technology and approaches used do not introduce significant advancements beyond existing work.
In my opinion, this paper appears to utilize existing methods for calculating the Hessian matrix and EIG, combining them with the $\log\det(\cdot)$ of the Fisher Information matrix to create a balanced objective function for exploration and improved self-localization. This approach is a relatively common technique in active SLAM and active localization fields.
The challenging part of gradient calculation of the 3DGS rendering model with respect to the pose, $\frac{\mathcal{D} f(\mathrm{x})}{\mathcal{D} \mathbf{x}}$, also relies on previously established methods. As such, the novelty and impact of the proposed approach feel somewhat limited in scope.

**2. Unclear Definitions**: The paper could benefit from clearer definitions of key terms and variables to improve readability. For example, the authors mention minimizing the difference between the image observation $\mathbf{y} $ and the 3DGS parameters $\mathbf{w}$ through the rendering model $f(\mathbf{x}, \mathbf{w})$, but they do not clarify detailed information of these variables such dimensionality.
Another example is the function $g(\cdot)$  in Equation (5), which lacks a clear definition, making it unclear how this function is formulated or what role it plays within the proposed framework.
Including additional details on variable dimensions and implementation specifics would significantly aid readers in understanding the methodology and reproducibility of the work.

**3. Computational Cost**: Although the authors claim that their approach functions as an online SLAM method, there is limited discussion on the computational efficiency or performance overhead of the proposed approach. Since online SLAM typically requires low latency and real-time processing capabilities, an experimental evaluation of computational efficiency would be beneficial. Including such results would strengthen the paper by validating its practical applicability and providing a more comprehensive view of its performance.

**Questions:**

See weakness.

---

### Official Review · Reviewer_21V3 · 2024-11-07

**Soundness:** 3
**Presentation:** 2
**Contribution:** 3
**Rating:** 6
**Confidence:** 4

**Summary:**

This paper proposed an active SLAM using 3DGS as scene representation. The core innovation is the active pose selection through balancing the information gain with respect to both location and the map based on Fisher Information matrix and the Cramer-Rao Lower Bound.

**Strengths:**

The introduction is reasonably motivated; the related work is adequately discussed; the methodology is theoretical, and the problem of active SLAM balancing the location and the map is modeled in a reasonable and innovative way; the experimental results prove the superior performance of the proposed method.

**Weaknesses:**

1.	The writing is often clumsy. You should proofread your document carefully.
2.	The good mapping and exploration performance exhibited by the proposed method may depend heavily on the utilization of the 3DGS, but the methodology section rarely deals with the description of the 3DGS integration. Following the current description, it seems that the proposed active pose selection method can be applied to any 3D representation, not just 3DGS.

**Questions:**

What is the method running time? If it is shown in the paper, it will help researchers to evaluate the practical value of the method.

---

### Note · Authors · 2024-11-15

I have read and agree with the venue's withdrawal policy on behalf of myself and my co-authors.